# Eliminating the need for manual segmentation to determine size and volume from MRI. A proof of concept on segmenting the lateral ventricles

**Fernando Yepes-Calderon** [ORCID][2,4]*, **J. Gordon McComb**[1,3]

**1** Division of Neurosurgery, Children's Hospital Los Angeles, Los Angeles, CA, United States of America, **2** Science-Based Platforms, Fort Pierce, Florida, United States of America, **3** Keck School of Medicine, University of Southern California, Los Angeles, CA, United States of America, **4** GYM Group SA, Cali, Colombia

* fernando.yepes@strategicbp.net

**Data Availability Statement:** Data is available at Zenodo (https://zenodo.org/) under DOI:10.5281/

## Abstract

Manual segmentation, which is tedious, time-consuming, and operator-dependent, is currently used as the gold standard to validate automatic and semiautomatic methods that quantify geometries from 2D and 3D MR images. This study examines the accuracy of manual segmentation and generalizes a strategy to eliminate its use. Trained individuals manually measured MR lateral ventricles images of normal and hydrocephalus infants from 1 month to 9.5 years of age. We created 3D-printed models of the lateral ventricles from the MRI studies and accurately estimated their volume by water displacement. MRI phantoms were made from the 3D models and images obtained. Using a previously developed artificial intelligence (AI) algorithm that employs four features extracted from the images, we estimated the ventricular volume of the phantom images. The algorithm was certified when discrepancies between the volumes—gold standards—yielded by the water displacement device and those measured by the automation were smaller than 2%. Then, we compared volumes after manual segmentation with those obtained with the certified automation. As determined by manual segmentation, lateral ventricular volume yielded an inter and intra-operator variation up to 50% and 48%, respectively, while manually segmenting saggital images generated errors up to 71%. These errors were determined by direct comparisons with the volumes yielded by the certified automation. The errors induced by manual segmentation are large enough to adversely affect decisions that may lead to less-than-optimal treatment; therefore, we suggest avoiding manual segmentation whenever possible.

## Introduction

Digital imaging has progressed to where it is utilized for an ever-increasing number of applications, many of which have become essential to modern society [1, 2]. Irrespective of the application, the visual information is subject to viewer interpretation requiring computational methods

zenodo.7654881. It can also be downloaded at URL https://doi.org/10.5281/zenodo.7654881.

**Funding:** This work was supported by the Rudi Schulte Research Institute, grant number RDP0000051 to JGM. The funder had no role in study design, data collection and analysis, decision to publish, or preparation of the manuscript.

**Competing interests:** The authors have declared that no competing interests exist.

to quantify [3–5]. In the quantifying processes, the method begins by determining if an identifiable boundary exists between a given structure and its surroundings within the field of view. If so, this allows segmentation to determine planar geometries (2D) and volume (3D). [6–8].

Cerebral ventricular volume is essential in diagnosing and treating neurological diseases, with hydrocephalus being one of the most common [9]. Serial MRI studies monitor ventricular size to gauge response to treatment and disease progression. Such monitoring is accomplished by visually comparing ventricular size from one set of MR images to another [10]. To more accurately determine if a change in ventricular volume has occurred, multiple semiautomatic and automatic techniques have been developed using manual segmentation as the gold standard for validating purposes [11, 12]. As manual segmentation is time-consuming, tedious, and operator-dependent, it is usually only done for research endeavors [13]. The question arises regarding how accurate manual segmentation is upon which semiautomatic and automatic programs have been developed for this purpose. This study examined the accuracy of manual segmentation for ventricular volume (3D) and compared it to a certified version of the Automatic Ventricular Volume Estimator (AVVE), a method we developed in [14]. The AVVE uses Support Vector Machine (SVM) to automatically classify the voxels belonging to volumes of interest. This statistical estimator receives four features extracted from the studied image and the ventricular masks as supervisory factors. When presented to the research community, the AVVE was validated using manually segmented masks, but in this delivery, the AVVE has been certified for accuracy using a reproducible pipeline. Then, with the certified AVVE, we measure and report the errors attained by human operators while segmenting the lateral ventricles.

## Materials and methods

The Fig 1 shows a generalization of the presented solution. The primary purpose is to create reliable gold standards that measure more accurately than manual segmentation and tune automatic or semiautomatic instruments in the measuring range.

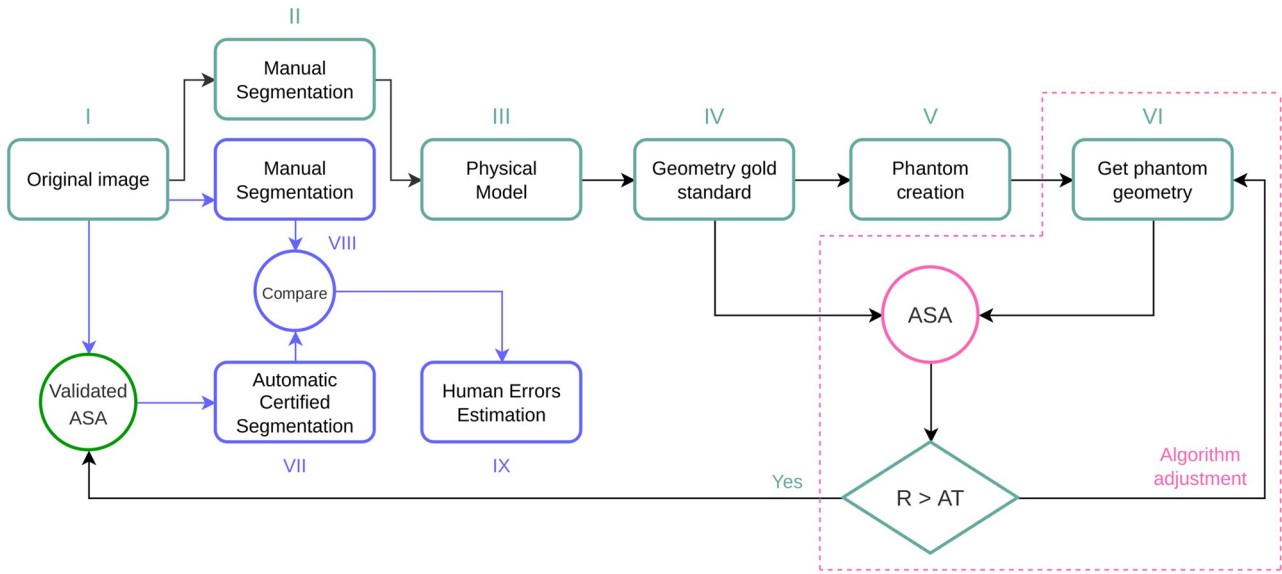

**Fig 1. General strategy to determine human errors in manual assessment.** The first pass over manual segmentation allows the creation of physical models serving as gold standards to tune an Automatic Segmentation Algorithm (ASA). The ASA is validated when yield volume read value differs (R) from the one given by the Water Displacement With the validated ASA the automatic and manual segmentations are compared to determine the human errors.

Since applications in medicine deal with individuals' health, the need for accuracy is high [15]. The methods presented in this document strictly comply with relevant guidelines and regulations. This study was approved by the Institutional Review Board of the Children's Hospital Los Angeles, which waived the requirement for informed consent because of the data's retrospective nature and use of de-identifying methods. Please, refer to IRB number CHLA-15-00161.

## Human errors in 3D measurements

Correctly estimating the volume of the brain's ventricles is crucial in diagnosing and monitoring hydrocephalus in infants and normal pressure hydrocephalus in adults [9, 16, 17]. Randomly selected images of the brain lateral ventricles were manually segmented to create 3D structure models. The volume of the 3D models was determined using an electronic device that reads water displacement (WD). The 3D models were used to create MRI phantoms scanned in a 3T Phillips device using isometric voxels of 1mm. From this moment, images are created from a 3D structure with a known volume. The volume of the 3D structure serves as a supervising factor that certifies the operation of the AVVE that performs segmentation of the ventricles using AI [14]. With the certified AI-based measurements, we determined errors introduced by human operators during manual segmentation. For this volume-target scenario, the general pipeline of Fig 1 turns into the one shown in Fig 2.

## Tunning process. The Brain-ventricles' phantoms

This Section points to the creation of the phantom shown in block V of Fig 2. The ventricles are segmented from T1 images. The resulting masks are saved in stereolithography (STL)

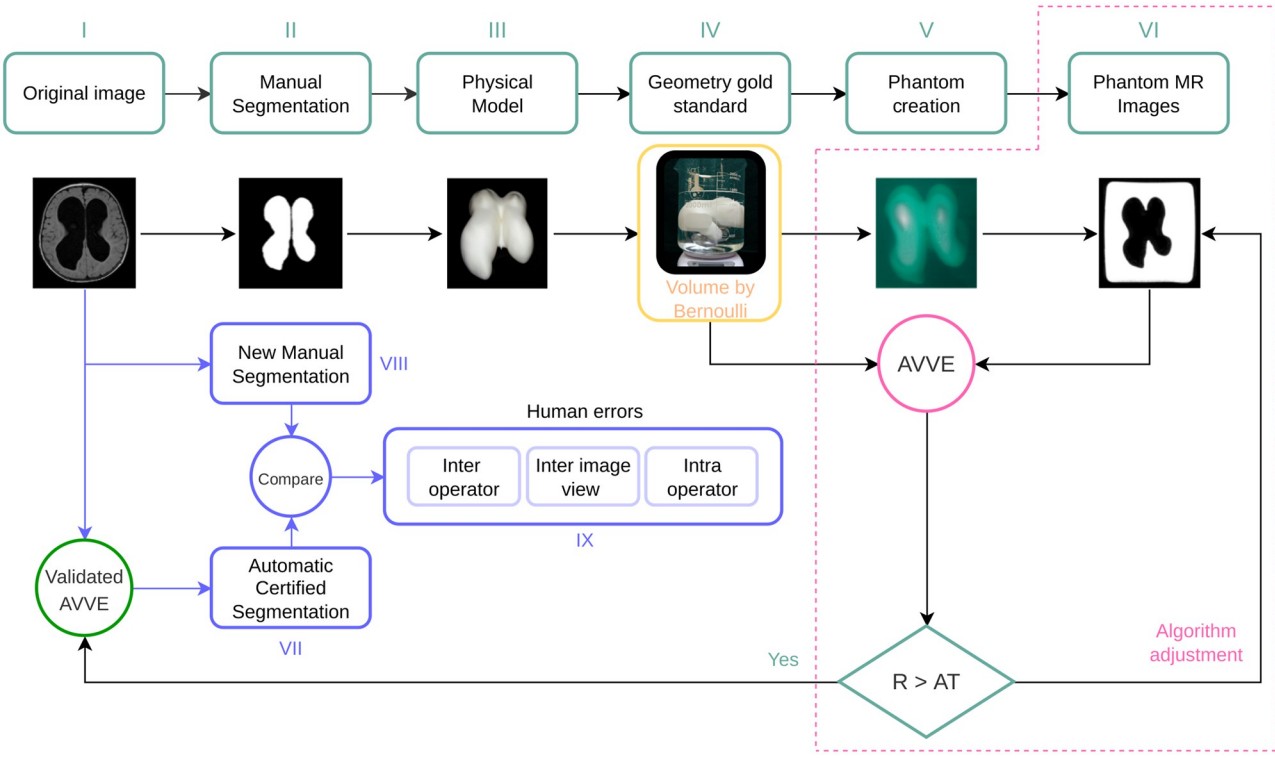

**Fig 2. We tested the automatic ventricular volume estimator (AVVE) algorithm [14] for accuracy using as a gold standard the volume of the 3D printed structures obtained by water displacement.**

format [18]. Next, the STL files are loaded in Cura [19] using a resolution of 0.1 mm on all axis. Then, the models are moved to gcode format [20] before printing in a Monoprice Ultimate 3D printer using 0.1mm of precision and 20% for structural filling. From this moment, a physical-measurable object exists with dimensions in the real world; however, its form is complex. From the physical models, MRI phantoms can be created. The process consists in suspending the volume in a solution jelly:water (1g:3ml). The inert material of the 3D model surrounded by the watery fixation creates the needed contrast on an MRI scanner from which images are obtained. From this point on, the volumes extracted from images can be fairly compared with those obtained by the water displaced with the physical model. The brain-ventricular models were extracted from templates created by healthy patients at ages [1, 6, 15, 24, 48, 66, 78, 96, 114] months old. Additionally, two hydrocephalus patients underwent the same process.

### Tunning process. The water displacement measuring device

The water displacement (WD) was chosen as the method to measure the irregular volumes of the 3D reproduced ventricles. The conceptual design of the device is shown in the Fig 3. The montage consists of a measuring [MR] and a sample recipient [SR], both hosting electrical

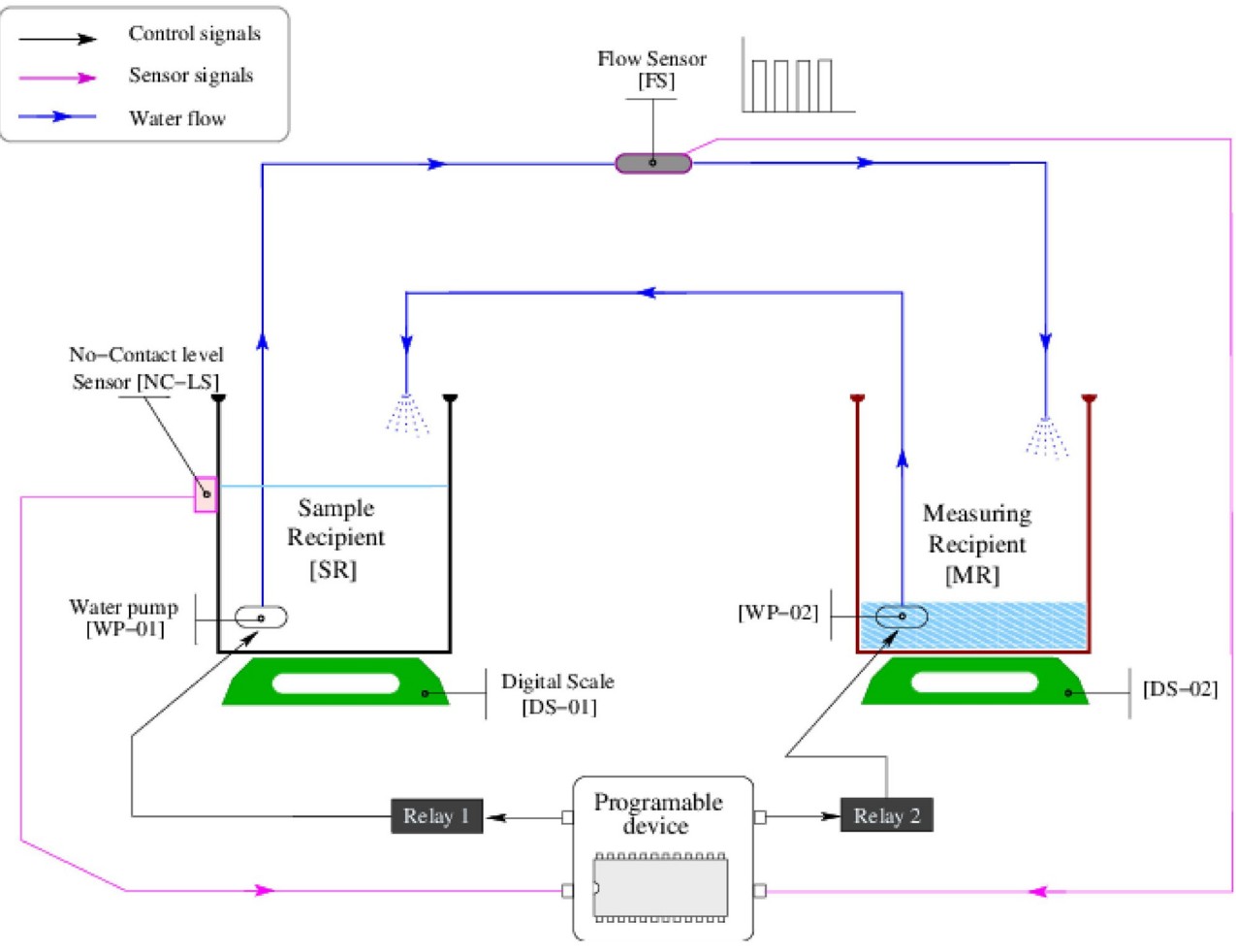

**Fig 3. Design of the water-displacement-measuring device.**

water pumps [WP-01] and [WP-02]. The recipients rest on digital scales [DS-01] and [DS-02] with a precision of 1 ml. The [SR] has a non-contact-level sensor [NC-LS], which works as a digital switch. The [NC-LS] is on when water reaches or exceeds the sensor level; it is off otherwise. The pumps are connected to two pipes so that depleting one recipient fills the other. The tube that drains the [SR] is connected to a flow sensor [FS] that produces pulses when the water moves. The [FS] is specified to read fluxes in the range of 0.1-3L/min. This hardware is controlled with a Beagle Blackbone [21] (Programmable device) that recovers logical transistor-to-transistor logic (TTL) signals in its sensor ports (magenta lines) and uses the control ports (black lines) to activate/deactivate the pumps over the residential power distribution (120V-60Hz) through transistorized power interfaces.

To start, the water level in [SR] is below the [NC-LS] sensor; thus, [NC-LS] sends a 0 through its sensor line. Then [WP-02] is activated to push water on [SR] until the water reaches the [NC-LS] level. At this moment, the programmable device will see a logic 1 in the [NC-LS] sensor line. Next, [WP-01] is activated to deplete water from [SR] to find the zero level. At that moment, the programmable device sees a zero in the [NC-LS] line. Then, the sample is submerged in [SR], raising the water level above the [NC-LS] sensor and forcing a logic 1 in the sensor line. Next, the [WP-01] is turned on, and the programmable device activates the pulse counting in the [FS] sensor line. The water pumping from [SR] will continue until the water level reaches zero. The volume of the displaced water is equal to the volume of the submerged object, and it will be captured by the pulsating pattern yielded by the [FS] sensor. Because the 3D volumes are built with gaps in their internal structure, sinkers are needed to eliminate the buoyancy.

### Tunning process. Estimating volume with artificial intelligence

Marbles of different sizes are utilized to accurately estimate the water's flux traversing the [FS] device. The marbles' volume is determined analytically by measuring the diameter (D) with a caliper with a precision of 0.1 mm and using $V = \frac{1*\pi*D^3}{6}$. The uncertainty of the device is estimated by measuring known volumes –the marbles– in the range of the studied ventricles. The uncertainty in each studied point is calculated by averaging five readings. The [FS] produces a pulsating signal where the proximity of the pulses is directly correlated with the flux (volume per time). Unfortunately, the pumps do not move the water at a constant rate; therefore, the pulsating pattern's first derivative yields signals with descending-exponential envelopes. Since the behavior of the pumps is challenging to characterize by analytical means, and such operational variability precludes accuracy in volume estimation, we tested several regression methods to predict volume from pulsating patterns. A regressor based on a neuronal network resulted in the best solution for the challenge. The pulse counting ($PuC_i$), the first 20 time slots produced by the first derivative of the pulsating pattern ($TS_i$), the time the system took to displace the water ($Twd_i$), and the amplitude $A_0f_i$ of the Fourier's DC components on each $TS_i$ are included in the input array $X_i$ (44x23). The output array $Y_i$ (44x1) contains the volumes—namely real volumes—per each formulation extracted analytically based on the caliper measurement. This data is publicly available at https://doi.org/10.5281/zenodo.7654881 [22].

Training/testing tasks were accomplished in a 3-folded exercise using randomly selected $TS_i$ arrays in a ratio 70%/30% among all pulsating samples. Data augmentation was accomplished by submerging several marbles together in [MR]. We avoid the need for padding in the matrix formulation—due to the different lengths of the pulsating patterns—by considering only the first 20 time slots in ($TS_i$). The decision not to use the whole ($TS_i$) was made after observing that the highest timing variability in the whole data set was due to the irregular pump's starting. The results section shows a Mean Absolute Error (MAE) metrics comparison

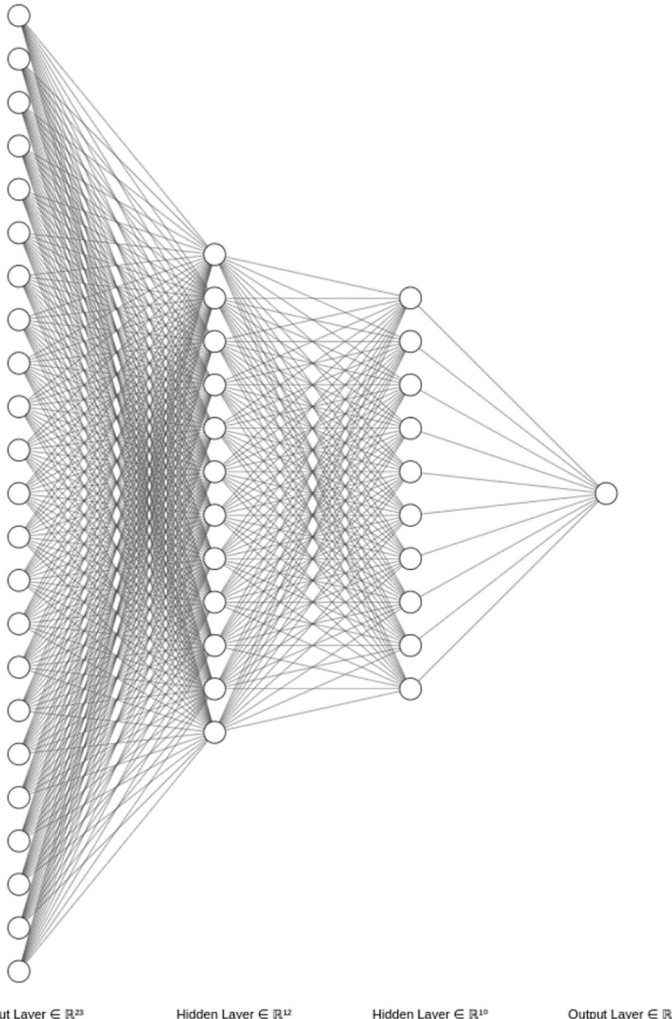

**Fig 4. Architecture of the used neuronal network.** The first layer has 23 nodes to receive the features in $X_s$, two hidden layers with 12 and 10 nodes, both layers with rectified linear unit (ReLu) activation function, and an output layer with a linear activation function that performs the regression. Mean Squared Error (MSE) drives the loss, the Adam algorithm is set as the optimizer, and the accuracy metric is performed by Mean Absolute Error (MAE).

among tested regressors, including linear regression, polynomial regression, Neuronal network with linear output layer, decision trees, and Random Forest. The Fig 4 presents the architecture of the used neuronal network that ultimately showed the best performance.

## Manual segmentation (MSeg) assisting software

The MSeg process involves tasks that are not related to tracing the outline of the ventricles but are also essential to accomplish the activity. These collateral activities refer to loading images, moving through slices, saving the mask obtained from the current slice, concatenating the masks, and saving the created volume.

Since our purpose is to qualify and quantify the segmentation process, the mentioned collateral activities are fully automated; therefore, the operator is forced to find and delineate the region of interest in every slice without distractions. Besides, we have accounted for operator

fatigue with timers that allow the operator to work for 30 minutes and force a 10 minutes rest before restarting the segmentation. These values were empirically chosen after receiving feedback from operators regarding optimal times to enforce concentration. The in-house-made software monitors some activities the operator performs and records timestamps before and after every action.

## Human assestments on clinical data

Four human operators have been trained to segment the lateral brain ventricles on MRI data available at the Children's Hospital Los Angeles. Although clinical imaging on children often yields low-quality images, the ventricles are among the most easily identifiable structures in the brain. Each operator is asked to separate the lateral brain ventricles three times in the three views for a total of 9 Mseg per subject. The gathered information is profiled and kept separated for subsequent analysis as follows:

**Inter-operator experiments.** The four operators perform segmentation on the axial images of patients in the same age range as those used for creating the gold standards. The volumes' mean values obtained among the experts are compared with those extracted with the water displacement device. Then, errors are computed.

**Inter-view experiments.** The operators perform segmentation on every view (axial, coronal, sagittal). The obtained volumes are computed for each view and compared with the volume yielded by the water displacement method to determine errors.

**Intra-operator experiments.** The operators are presented with the task of segmenting every available structure in the axial view three times. Since the assisting software controls how the images are delivered, operators are never conferred with the same subject consecutively, so learning is avoided. The mean value of the obtained volume is computed with the volumes yielded by the water displacement method to obtain the difference.

## Results

### Water displacement device and yielded data

The differences in the timing profiles described by the Gaussian statistics in Table 1 suggest an irregular operation in the pumping device that tends to stabilize itself when the pump is operative for extended periods. The construction of the flow sensor forces a pulsation pattern that does not vary its duty cycle (50%) but its frequency, justifying the use of a feature extracted from the Fourier in the AI-based volume prediction tasks.

**Table 1. Marbles' volumes and time slot statistics per experiment.** Ma, Vol, Max, Min, and Std stand for Marble, Volume, Maximum, Minimum and Standard deviation, respectively.

| Count (pulses) | Model | Real Vol (ml) | Max time | Mean time | Min time | Std time |
|---|---|---|---|---|---|---|
| 58 | Ma2 | 8.0 ± 1.0 | 0.037554 | 0.030398 | 0.025835 | 0.000993 |
| 61 | Ma1 | 8.1 ± 1.0 | 0.037157 | 0.030073 | 0.027922 | 0.000779 |
| 63 | Ma5 | 7.9 ± 1.0 | 0.037871 | 0.030483 | 0.027922 | 0.000795 |
| 65 | Ma4 | 7.6 ± 0.9 | 0.037233 | 0.031458 | 0.027922 | 0.001300 |
| 66 | Ma3 | 7.3 ± 0.9 | 0.037157 | 0.031326 | 0.027922 | 0.000707 |
| 120 | Ma6 | 16.1 ± 1.4 | 0.033562 | 0.030225 | 0.028124 | 0.000726 |
| 184 | Ma7 | 23.4 ± 1.6 | 0.037157 | 0.031093 | 0.026125 | 0.000963 |
| 252 | Ma7 | 35.6 ± 0.9 | 0.033232 | 0.029899 | 0.025774 | 0.000930 |
| 445 | Ma8 | 60.1 ± 0.9 | 0.034960 | 0.030765 | 0.026517 | 0.000837 |

**Table 2. Mean Absolute Errors (MAE) in three folded exercises aiming to predict the volume from features derived from pulsating patterns.** The MAE records are presented in $mm^3$. The term reg stands for regression.

| Folding | Regression model (23 features, 44 formulations, split 0.3) | | | | |
|---|---|---|---|---|---|
| | Linear reg. | Polynomial reg. | Neuronal Network | Decision tree | Random forest |
| | MAE | MAE | MAE | MAE | MAE |
| 1 | 1578.54 | 3260.24 | 129.99 | 1908.61 | 578.12 |
| 2 | 1876.58 | 4125.73 | 79.71 | 1155.41 | 411.65 |
| 3 | 1599.33 | 3518.47 | 85.81 | 1196.66 | 388.96 |
| Average | 1684.81 | 3694.81 | 98.50 | 1420.22 | 459.57 |

The Table 1 is a record of the pulses generated by some of the marbles used in the WD-device's tuning process. The pulse counting sorts the data; however, the order is not kept in the column Real Vol, empowering the thesis of the pump's unstable behavior, which is corroborated by timings registered in the same table.

### Tunning process. Volume estimation through artificial intelligence

Five regression strategies were tested to convert the pulsating patterns to volume after submerging marbles in the water displacement device. Table 2 compares the MAE for each regressor in a three-fold exercise.

In Table 2, the polynomial regression is configured with a 2nd order degree. The Neuronal network uses the MSE to calculate loss, the Adam optimizer, and the activation functions per layer were: relu, relu, relu and linear. The decision tree and random forest methods were configured as regressors, and the random forest used ten estimators.

The smallest marble's volume read with a caliper yielded an analytical value of 7329 mm3, and the worst obtained MAE is 129.99 mm3 which is 1.77% (below the 2% of tolerance) of the smallest measured volume. Therefore, the WD is certified to measure volumes by water displacement with high precision.

### Estimating human errors in manual segmentation

Once the WD device is tuned, it is possible to measure the 3D structures' volume from medical images accurately.

The Fig 5 showing errors as percentual differences concerning the gold standard volumes complements Table 3.

The circle's center is the radar plots' zero error point. The errors are presented as a percent of the real value provided by the water displacement method (i.e., gold standard). The inter-

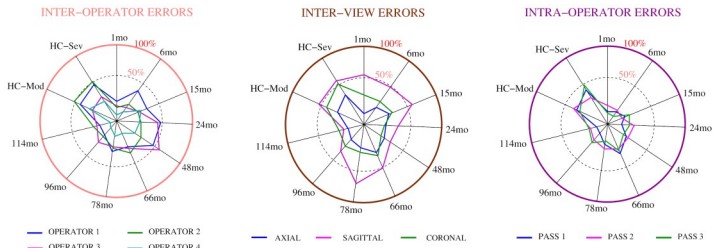

**Fig 5. Graphical results of the performed experiments.** The errors are presented as percentual variations from the gold standard stated by the water displacement device. The abbreviations HC, mo, Mod and Sev stand for hydrocephalus, month, moderate and severe, respectively.

**Table 3. The AVVE value column holds the volumes obtained by the certified AVVE on clinical images.** The experts are asked to perform segmentations on the same subjects measured by the certified AVVE. The mean values obtained by the experts on each subject are registered in this table. Using the mean values obtained from several operators is a strategy often used to validate the accuracy of automatic and semiautomatic segmentation tools. The abbreviations HC, mo, mod, and sev stand for hydrocephalus, month, moderate and severe, respectively.

| PATIENT AGE (mo) | AVVE VALUE (ml) | INTER − OP EXPs n = 4 | | INTER − VIEW EXPs n = 3 | | INTRA − OP EXPs n = 3 | |
|---|---|---|---|---|---|---|---|
| | | Mean (ml) | Error (ml) | Mean (ml) | Error (ml) | Mean (ml) | Error (ml) |
| 1 | 3.4 ± 0.2 | 3.9 ± 0.3 | 0.5 ± 0.4 | 4.42 ± 0.7 | 1.0 ± 0.7 | 4.0 ± 0.2 | 0.6 ± 0.3 |
| 6 | 7.3 ± 0.2 | 8.9 ± 0.1 | 1.60 ± 0.2 | 9.6 ± 0.1 | 2.3 ± 0.2 | 8.4 ± 0.1 | 1.1 ± 0.2 |
| 15 | 10.8 ± 0.2 | 13.6 ± 0.7 | 2.8 ± 0.7 | 14.6 ± 1.7 | 3.8 ± 1.7 | 12.5 ± 0.6 | 1.7 ± 0.6 |
| 24 | 10.5 ± 0.2 | 14.2 ± 1.4 | 3.6 ± 1.4 | 13.2 ± 1.1 | 2.7 ± 1.1 | 12.0 ± 0.5 | 1.5 ± 0.5 |
| 48 | 19.8 ± 0.2 | 17.9 ± 2.6 | 1.9 ± 2.6 | 16.7 ± 0.9 | 3.1 ± 0.9 | 16.3 ± 0.3 | 3.5 ± 0.4 |
| 66 | 8.0 ± 0.2 | 9.8 ± 0.8 | 1.8 ± 0.8 | 11.1 ± 0.9 | 3.1 ± 0.9 | 9.9 ± 0.1 | 1.9 ± 0.2 |
| 78 | 11.5 ± 0.2 | 13.9 ± 1.1 | 2.4 ± 1.1 | 16.3 ± 3.0 | 4.8 ± 3.0 | 14.0 ± 0.3 | 2.5 ± 0.4 |
| 96 | 11.0 ± 0.2 | 12.7 ± 1.0 | 1.7 ± 1.0 | 14.3 ± 0.9 | 3.3 ± 0.9 | 13.0 ± 0.7 | 2.0 ± 0.7 |
| 114 | 19.7 ± 0.3 | 23.3 ± 1.8 | 3.6 ± 1.8 | 23.7 ± 0.9 | 4.0 ± 0.9 | 22.5 ± 0.9 | 2.8 ± 0.9 |
| HC-mod (72) | 88.4 ± 0.9 | 113.5 ± 18.4 | 25.0 ± 18.4 | 119.6 ± 21.1 | 31.2 ± 21.1 | 107.2 ± 7.8 | 18.8 ± 7.8 |
| HC-Sev (80) | 115.9 ± 1.0 | 152.8 ± 21.3 | 36.9 ± 21.3 | 161.2 ± 33.3 | 45.3 ± 33.3 | 136.9 ± 18.4 | 21.0 ± 18.4 |

operator measurements introduced errors up to 50% concerning the water displacement standard, and the more significant volumes tended to be more challenging to measure. Regarding the plane, the operators were more accurate in segmenting the ventricles when working in the axial view. Segmenting the sagittal plane generated the most significant errors reaching differences up to 71% with respect to the water displacement standard. The Intra-operator variability reached 48%, and the most extensive volumes presented the highest challenges to the human operators.

## Discussion

A significant number of algorithms—many of them employing AI—have been developed to generate semiautomatic and automatic determinations of volume and shape. Scientists employ different imaging modalities to quantify geometries and later judge accuracy against manual segmentation as the gold standard. Several such studies in medicine have assumed that manual segmentation is a reliable validator [23–30]. In this report we demonstrate that manual assessments are not that accurate. Moreover, we provided insights about a highly accurate technique with a proof of concept that eliminates the need to use inaccurate, tedious, time-consuming, and operator-dependent manual segmentation.

The problem of validating automatic and semiautomatic tools with manual assessments in medicine has been underrated. Nevertheless, some authors have recently spoken out about the inconsistency of using unstable manual segmentation as a grand truth and proposed to believe in the AI-based machine's capacity to learn and be reproducible [31] for accomplishing tasks with precision. The authors in [31] justified their efforts with a 10% discrepancy between operators in a multiple-sclerosis framework while segmenting brain structures. However, reporting the differences between operators obviates the target and, thus, precision. In other words, both operators could report the same and be remote from the real numbers. Losing the target is a natural result when we lack an objective gold standard. This missing part propagates the inaccuracy to the AI machine performing the segmentation. Has it obtained the correct numbers? How can we ensure that? We can not compare our findings with anything reported before because we propose the creation of reliable gold standards, something missing in the 8.880

entries displayed by google scholar after the search string "Segmentation algorithms in medical imaging" only in 2023.

In the presented scenario, we created 3D printed models derived from MR images that mimic the lateral ventricles and very precisely measured the 3D models' volume with a water displacement technique. The 3D models were placed in a gel, and MR images were obtained. The images extracted from the phantoms were fed to an AI-based algorithm tuned until the volumes were congruent to those obtained by water displacement. The next step, currently in development, is to incorporate the algorithm into MR scanners so that all subsequent ventricular volumes can be accurately and automatically be determined with a numerical value that will be included with each radiology report.

Similarly, for planar measurements in medicine (study not reported in this document), we created multiple printed rings to mirror the mean HC at various ages along the x-axis of the Nellhaus chart for head growth. The 3D printed rings (arbitrarily made 0.5 cm thick to have substance) were used to create MR phantoms in a manner equivalent to that of the ventricular models. Measurements of the outside diameter of 3D rings tuned the automatic image-based algorithm that determines maximum HC [32]. The algorithm to accomplish automatic and accurate measurement of maximum HC is currently being added to the MR scanners at our hospital. We are not advocating that pediatric patients undergo MR imaging to obtain maximum HC but to utilize images incorporated in the hospital database acquired for clinically indicated purposes. Human errors in planar images or measurements directly performed on the body, such as the maximum HC, can be accurately estimated using the model proposed in Fig 1. We are currently working on this development and will report the referred errors subsequently.

The Picture Archiving and Communications System (PACS) [33] is currently the standard platform to manage medical images but lacks analytical and quantification capabilities. Staying within the PACS, we have developed automatic methods to retrieve the medical data and access it at the voxel level, decrypted and uncompressed enabling analytical procedures to be applied to the data while not perturbing the system's daily operations.

The Health Insurance Portability and Accountability Act (HIPAA) [34], a federal law enacted in 1996 to protect patients' health information, mandates that such information cannot be disclosed without a patient's consent. Data transferred out of the PACS is identifiable and, thus, is subject to all the requirements of HIPAA. By eliminating manual segmentation, we add reliability to the whole automating pipeline and assure that our methods are HIPAA compliant, eliminating the need for patient or Institutional Review Board (IRB) approvals. Doing so also makes it much easier to monitor a given patient over time and compare such a patient with other patients included in a defined database. The presented automation can be expanded by including multiple institutional sites that favor implementing AI in medicine, as we displayed in patent US20200273551A1 [35].

## Conclusion

Manual segmentation is not recommended to derive quantitative assessments from medical images nor to validate automatic or semiautomatic methods based on such a technique since the results of the Mseg are variable and do not provide a mechanism to determine the accuracy of the results. The errors induced are large enough to adversely affect decisions that may lead to less-than-optimal treatment.

The results yielded by automation can be made to be reproducible. The inaccuracies introduced by machines are known as systematic errors, and those discrepancies can be corrected if a reliable gold standard is utilized. The authors recommend that automation using quantifiable

gold standards be used to determine size and volume from medical images with manual segmentation be eliminated whenever possible.

## Author Contributions

**Conceptualization:** Fernando Yepes-Calderon, J. Gordon McComb.

**Data curation:** Fernando Yepes-Calderon.

**Formal analysis:** Fernando Yepes-Calderon.

**Funding acquisition:** J. Gordon McComb.

**Investigation:** Fernando Yepes-Calderon, J. Gordon McComb.

**Methodology:** Fernando Yepes-Calderon, J. Gordon McComb.

**Project administration:** J. Gordon McComb.

**Resources:** J. Gordon McComb.

**Software:** Fernando Yepes-Calderon.

**Supervision:** J. Gordon McComb.

**Validation:** Fernando Yepes-Calderon.

**Visualization:** Fernando Yepes-Calderon.

**Writing – original draft:** Fernando Yepes-Calderon.

**Writing – review & editing:** Fernando Yepes-Calderon, J. Gordon McComb.

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
