## [Decision Letter · Decision Letter 0]

14 Dec 2022

PONE-D-22-26535Eliminating the Need for Manual Segmentation to Determine Size and Volume from MRIPLOS ONE

Dear Dr. Yepes-Calderon,

Thank you for submitting your manuscript to PLOS ONE. After careful consideration, we feel that it has merit but does not fully meet PLOS ONE’s publication criteria as it currently stands. Therefore, we invite you to submit a revised version of the manuscript that addresses the points raised during the review process.

We look forward to receiving your revised manuscript.

Kind regards,

Kumaradevan Punithakumar

Academic Editor

PLOS ONE

Journal Requirements:

Reviewers' comments:

Reviewer's Responses to Questions

**Comments to the Author**

1. Is the manuscript technically sound, and do the data support the conclusions?

Reviewer #1: Yes

Reviewer #2: Yes

2. Has the statistical analysis been performed appropriately and rigorously? 

Reviewer #1: Yes

Reviewer #2: Yes

3. Have the authors made all data underlying the findings in their manuscript fully available?

Reviewer #1: No

Reviewer #2: Yes

4. Is the manuscript presented in an intelligible fashion and written in standard English?

Reviewer #1: Yes

Reviewer #2: Yes

5. Review Comments to the Author

Reviewer #1: Table 2 shows the performance of the neuronal network model. However, an additional table is also needed that should compare the neuronal network model presented in this paper with other current state-of-the-art models. I recommend showing the accuracy, recall, F1-score, and precision of this neuronal network model and the other compared previous models.

Reviewer #2: Dear authors, it is an interesting paper about segmentation from MRI.

I have the following suggestions:

Title: please change for: Eliminating the need for manual segmentation to determine the size and volume of lateral ventricles images from MRI

Abstract: at the first Page of the paper please review the numbers: 2\\%, etc

The conclusions of the abstract is not so clear as the conclusion of the paper. Please, review the conclusion of the abstract.

Introduction: please explain in few words the reference 14.

Methods: Please include the abbreviations of the Figures and Tables at the legend after each Figure and Table.

Results: the interoperator measurements errors are up to 50%. What are the explanations for such number and other high Numbers errors?

Discussion: please compare the findings of this study with other studies of automation measurements in MRI.

6. PLOS authors have the option to publish the peer review history of their article (what does this mean?). If published, this will include your full peer review and any attached files.

Reviewer #1: **Yes: **Ayush Goyal

Reviewer #2: **Yes: **Vera Maria Cury Salemi

---

## [Author Response · Author response to Decision Letter 0]

3 Mar 2023

Dear reviewers,

We highly appreciate all your suggestions and comments. We have used your feedback to improve the form and content of our document. The responses to your questions are registered below your statements. The actions taken are recorded in this document and performed in the manuscript; changes and additions are marked in bold font.

Again, thanks for your time.

Reviewer 1.

Table 2 shows the performance of the neuronal network model. However, an additional table is also needed that should compare the neuronal network model presented in this paper with other current state-of-the-art models. I recommend showing the accuracy, recall, F1-score, and precision of this neuronal network model and the other compared previous models.

R/ Thanks for these suggestions. The accuracy, recall, and F1 score are metrics for classification solutions. They all use the terms of the confusion matrix, and their domain is the discrete world. In the context of this work, translating the pulses to a volume is better framed by regressors since the output must be a number in the continuous domain. In this case, the Mean Absolute Error (MAE) is preferred. The Mean Squared Error (MSE) is also used, but physicians used to have a better sense of accuracy with the MAE since it does not involve quadratic operators.

We followed your suggestion and tested other state-of-the-art AI approaches. They are:

-Linear regression

-Polynomial regression

-Neuronal Network

-Decision tree

-Random forest

The MAE of the listed regression methods is presented in the results section of the paper.

Although these methods are comparable by MAE, the comparison remains unfair since there is space to configure them to improve their performance independently. It would be interesting to see how the simplistic linear and polynomial regressions would behave when optimized. The presented development will be enhanced if the simplistic approaches reach a good accuracy (low MAE) because it would make the embedded hardware run faster. 

The optimization of other methods was not executed due to the excellent performance of the neuronal network. Recall that the smallest volume read with a caliper yielded an analytical volume of 7329 mm3, and the worst obtained MAE is 129.99 mm3 which is 1.77% of the smallest measured volume. Therefore, the WD is certified to measure volumes by water displacement with high precision.

Reviewer 2.

- Dear authors, it is an interesting paper about segmentation from MRI. I have the following suggestions:

Title: please change for: Eliminating the need for manual segmentation to determine the size and volume of lateral ventricles images from MRI

R/ Thanks for your suggestion. We have proposed this method as a general mechanism to determine errors attained by manual segmentation and consequently discourage its use in medicine. Segmenting the ventricles is proof of concept, but we could have performed the same in any other body part. We are interested in keeping it general. We propose to set the title as Eliminating the Need for Manual Segmentation to Determine Size and Volume from MRI. A proof of concept on Segmenting the Brain Lateral Ventricles. 

Abstract: at the first Page of the paper please review the numbers: 2\\%

R/ 2% of a discrepancy between the AVVE automation and the gold standard was the limit we proposed to validate the operation of the AVVE. Therefore, the number is correct.

The other percentages are the highest discrepancies between the volumes measured by human operators and the certified automation, and they are also correct. 

The errors are a percentages of the volume measured with a certified tool. 

The conclusions of the abstract is not so clear as the conclusion of the paper. Please, review the conclusion of the abstract.

R/ That is true. We have changed the last paragraph of the abstract, and now it reads as follows.

"

The errors induced are large enough to adversely affect decisions that may lead to less-than-optimal treatments; therefore, we suggest avoiding manual segmentation whenever possible.

Introduction: please explain in few words the reference 14.

R/ We have written a synopsis of the method, which appears in bold fonts in the new version of the article. It reads as follows:

"

This study examined the accuracy of manual segmentation for ventricular volume (3D) and compared it to a certified version of the Automatic Ventricular Volume Estimator (AVVE), a method we developed in [14]. The AVVE uses Support Vector Machine (SVM) to classify the voxels belonging to volumes of interest automatically. This statistical estimator receives four features extracted from the studied image and the ventricular masks as a supervisory factor. When presented to the research community, the AVVE was validated using manually segmented masks, but in this delivery, the AVVE has been certified for accuracy using a reproducible pipeline. Then, with the certified AVVE, we measure and report the errors attained by human operators while segmenting the lateral ventricles.

"

Methods: Please include the abbreviations of the Figures and Tables at the legend after each Figure and Table.

R/ We have expanded all abbreviations used in figures and tables. Please observe the bold fonts in every updated caption.

Results: the interoperator measurements errors are up to 50%. What are the explanations for such number and other high Numbers errors?

R/ Such a considerable error rate is often read in significant volumes where there is more chance to make mistakes due to more extended boundaries. Also, larger structures are more affected by partial volume effects. In general, regardless of the errors' nature (big or small) concerning a gold standard in this artisan activity can be only explained by human factors. 

Discussion: please compare the findings of this study with other studies of automation measurements in MRI.

R/ The article aims to quantify and report human errors during segmentation tasks. We selected the lateral ventricles (LV) because our team has significant experience with these structures. The LV creates a good contrast in MRI and CT, even in low-quality acquisitions, facilitating the reproducibility of our methods. We could not find any other paper reporting manual segmentation errors that referred to a reliable gold standard while measuring LV in children or a different structure in any other type of subject. 

Papers report LV volumes [Melhem et al. 2000; Sarı et al. 2015] but their methods use manual segmentation or indirect mechanism such as the Evans' index; therefore, there is no shared space for comparison. Some reported volumes in [Melhem et al. 2000] may match the age ranges that we register in this manuscript; however, their patients have a brain malformation different from hydrocephalus, which is the only abnormality we report. Other authors declare VL volumes of various pathologies [Del Re et al. 2016; Ertekin et al. 2016; Turner,Greenspan & van Erp 2016], based on manual segmentation.

The problem of validating automatic and semi-automatic tools with manual assessments in medicine has been underrated. Nevertheless, some authors have recently spoken out about the inconsistency of using unstable manual segmentation as a grand truth and proposed to believe in the machine's capacity to learn and be reproducible [Zhang et al. 2020] for accomplishing tasks with precision. [Zhang et al. 2020] justified their efforts with a 10% discrepancy between operators in a multiple-sclerosis framework while segmenting brain structures. However, reporting the differences between operators obviates the target and, thus, precision. In other words, both operators could be in the same numbers and far away from the real numbers. Losing the target is a natural result when we lack an objective gold standard. This missing part propagates the hesitations to the scenario where the artificial intelligence machine performs the segmentation. Is it obtained the correct numbers? How can we ensure that? Still, we can not compare our findings with anything reported before because we propose the creation of a gold standard, something missing in the 8.880 entries displayed by google scholar after the search string "Segmentation algorithms in medical imaging" only in 2023.

We will include this explanation and references in the article's discussion section.

References

Del Re, E. C.; Konishi, J.; Bouix, S.; Blokland, G. A.; Mesholam-Gately, R. I.; Goldstein, J.; Kubicki, M.; Wojcik, J.; Pasternak, O.; Seidman, L. J. and others (2016). Enlarged lateral ventricles inversely correlate with reduced corpus callosum central volume in first episode schizophrenia: association with functional measures, Brain imaging and behavior 10 : 1264-1273.

Ertekin, T.; Acer, N.; Köseoğlu, E.; Zararsız, G.; Sönmez, A.; Gümüş, K. and Kurtoğlu, E. (2016). Total intracranial and lateral ventricle volumes measurement in Alzheimer’s disease: A methodological study, Journal of Clinical Neuroscience 34 : 133-139.

Melhem, E. R.; Hoon Jr, A. H.; Ferrucci Jr, J. T.; Quinn, C. B.; Reinhardt, E. M.; Demetrides, S. W.; Freeman, B. M. and Johnston, M. V. (2000). Periventricular leukomalacia: relationship between lateral ventricular volume on brain MR images and severity of cognitive and motor impairment, Radiology 214 : 199-204.

Sarı, E.; Sarı, S.; Akgün, V.; Özcan, E.; Ìnce, S.; Babacan, O.; Saldır, M.; Açıkel, C.; Başbozkurt, G.; Yeşilkaya, Ş. and others (2015). Measures of ventricles and evans' index: from neonate to adolescent, Pediatric neurosurgery 50 : 12-17.

Turner, A. H.; Greenspan, K. S. and van Erp, T. G. (2016). Pallidum and lateral ventricle volume enlargement in autism spectrum disorder, Psychiatry Research: Neuroimaging 252 : 40-45.

Zhang, L.; Tanno, R.; Xu, M.-C.; Jin, C.; Jacob, J.; Cicarrelli, O.; Barkhof, F. and Alexander, D. (2020). Disentangling human error from ground truth in segmentation of medical images, Advances in Neural Information Processing Systems 33 : 15750-15762.

---

## [Decision Letter · Decision Letter 1]

24 Apr 2023

Eliminating the Need for Manual Segmentation to Determine Size and Volume from MRI. A proof of concept on segmenting the lateral ventricles.

PONE-D-22-26535R1

Dear Dr. Yepes-Calderon,

We’re pleased to inform you that your manuscript has been judged scientifically suitable for publication and will be formally accepted for publication once it meets all outstanding technical requirements.

Kind regards,

Kumaradevan Punithakumar

Academic Editor

PLOS ONE

Additional Editor Comments (optional):

Reviewers' comments:

Reviewer's Responses to Questions

**Comments to the Author**

1. If the authors have adequately addressed your comments raised in a previous round of review and you feel that this manuscript is now acceptable for publication, you may indicate that here to bypass the “Comments to the Author” section, enter your conflict of interest statement in the “Confidential to Editor” section, and submit your "Accept" recommendation.

Reviewer #1: (No Response)

Reviewer #2: All comments have been addressed

2. Is the manuscript technically sound, and do the data support the conclusions?

Reviewer #1: Yes

Reviewer #2: Yes

3. Has the statistical analysis been performed appropriately and rigorously? 

Reviewer #1: Yes

Reviewer #2: Yes

4. Have the authors made all data underlying the findings in their manuscript fully available?

Reviewer #1: Yes

Reviewer #2: Yes

5. Is the manuscript presented in an intelligible fashion and written in standard English?

Reviewer #1: Yes

Reviewer #2: Yes

6. Review Comments to the Author

Reviewer #1: From the paper's title "Eliminating the Need for Manual Segmentation to Determine Size and Volume from MRI. A proof of concept on segmenting the lateral ventricles", it seemed the authors would have presented a novel algorithm for segmentiing ventricles or proposed a novel technique or at-least an improved method. However, the authors did not present or propose a novel method. They only compared existing methods in Table 2. Hence, this paper has limited originality and novelty. It seems that this paper is only showing results of existing methods.

Reviewer #2: Dear authors, the paper about segmentation for determination of LV size and volume.

My questions were answered and I don’t have suggestions.

7. PLOS authors have the option to publish the peer review history of their article (what does this mean?). If published, this will include your full peer review and any attached files.

Reviewer #1: No

Reviewer #2: No

---

## [Editor Report · Acceptance letter]

2 May 2023

PONE-D-22-26535R1 

Eliminating the Need for Manual Segmentation to Determine Size and Volume from MRI. A proof of concept on segmenting the lateral ventricles. 

Dear Dr. Yepes-Calderon:

I'm pleased to inform you that your manuscript has been deemed suitable for publication in PLOS ONE. Congratulations! Your manuscript is now with our production department. 

Kind regards, 

on behalf of

Professor Kumaradevan Punithakumar 

Academic Editor

PLOS ONE